# Performance Evaluation of Stormwater Management Systems and Its Impact on Development Costing

**Farjana Akhter \*, Guna A. Hewa, Faisal Ahammed, Baden Myers and John R. Argue**

School of Natural and Built Environments, University of South Australia, Mawson Lakes 5095, Australia; guna.hewa@unisa.edu.au (G.A.H.); faisalahammed.ahammed@unisa.edu.au (F.A.); baden.myers@unisa.edu.au (B.M.); john.argue@unisa.edu.au (J.R.A.)

\* Correspondence: akhfy002@mymail.unisa.edu.au; Tel.: +610470257234

**Abstract:** The contribution of this paper is a comparison of the installation cost of a conventional drainage system consisting of a network of pits and pipes, with that of a hybrid drainage system comprising a network of pits and pipes, supported by allotment scale infiltration measures in a modern greenfield residential development. The case study site is located in Pipers Crest, near Strathalbyn, South Australia. This as-built site consists of 56 allotments, 42 pits (hence 42 sub-catchments), one detention basin and over 1000 m of drainage pipes. In this study, conventional and hybrid (combination of conventional and Water Sensitive Urban Design, WSUD systems) drainage systems were designed to convey minor storm events of 10% annual exceedance probability (AEP), and checked for major storm events of 5% AEP, using the DRAINS model and/or source control principles. The installation costs of the conventional and hybrid drainage systems were estimated and compared based upon cost estimates derived from Australian literature. The results of the study indicate that satisfactory drainage was possible using the conventional or hybrid system when the two systems were designed to have outflow not exceeding the pre-developed flow. The hybrid drainage system requires smaller pipe sizes compared to the conventional system. Also, the size of the detention basin and maximum outflow rate of the hybrid system were smaller than those for the conventionally drained site. The installation cost of the hybrid drainage system was 18% less than that of the conventional drainage system when the objective was to accommodate 10% and 5% AEP storms.

**Keywords:** urban drainage; DRAINS model; source control; WSUD; costing

## 1. Introduction

### 1.1. Backgraound

Urban hydrology is a part of land hydrology, defined as an interdisciplinary science of water, investigating biochemical and physical changes to the hydrological processes and their impacts on an urban catchment [1–4]. Bajracharya et al. [5] stated that urbanization is a universal trend of the twenty-first century. Population growth and the movement of people from rural to urban settings cause land use changes in the form of urbanization [6]. Around 55% of the global population currently lives in urban areas, and over 500 of the world's cities have more than one million residents [1,7]. Australia has a population of 25.1 million, where 67% of total population lives in eight major urban developed cities [8,9]. Land use changes affect storm runoff characteristics; significant increases in runoff volume are caused by increased impervious areas such as roofs, roads, parking lots, footpaths and other imperviousness in the urbanized landscape [4,5,10,11]. Urbanization can also lead to a reduction in infiltration, decreases in groundwater recharge and deteriorating water quality in the drainage system [5,12–14].

As a result, urbanization increases the possibility of regular flooding, can compromise the effect of existing drainage facilities, increase the erosion of natural streams and can lead to water quality deterioration in receiving water bodies [1,5,15,16]. So, urban hydrology has been developed as a specialized field to explore these problems [17].

Urbanization can also affect local wind and climate systems, leading to increase in runoff and evaporation. Local effects can include reduced vegetation and reduced soil moisture, resulting in temperature increases [18,19]. Changes in vegetation cover and radiation can have a significant effect on precipitation over a local urban area and its surroundings [20].

To overcome problems associated with urbanization, innovation in urban stormwater management has become vital for urban planning, design and management [21]. Prior to the 1980s, stormwater runoff was considered as a waste product requiring quick disposal to receiving water bodies [22,23]. The main objective of a conventional drainage system is to collect and dispose of stormwater runoff as quickly as possible from residential and commercial areas to nearby receiving water bodies to avoid flooding [24–28].

Conventional systems provide no opportunity to save stormwater as a resource because the objective is to remove runoff within as short a time as possible [29–33]. A range of stormwater management approaches including conventional drainage systems, water sensitive urban design (WSUD) systems and Best Management Practices (BMPs), have been developed to mitigate flooding and water quality problems over the last few decades [1,34]. These WSUD concepts were introduced in Australia in the 1990s [22,35]. A component of the WSUD approach to water management is "source control". The main aim of source control is to hold the rainwater where it falls, thereby minimizing the negative impact of changes to the water cycle caused by urbanization. Applying source control and other WSUD techniques helps not only to manage the quantity of urban runoff, but also improves water quality before it reaches the receiving water bodies [34,36,37]. The concept of WSUD is known by different names in other countries; for example: Low Impact Development (LID), Low Impact Urban Development and Design (LIUDD), Sustainable Drainage Systems (SuDS) and Experimental Sewer System (ESS) [1,24,38].

*1.2. Aims of the Paper*

The first task of the project was to design the elements of a conventional storm drainage system for the 56-lot greenfield development case study at Pipers Crest in South Australia. This was followed by the design of on-site elements of the development considered from the water-sensitive perspective, as well as a supporting street drainage system, including pipes and pits. The result was a "hybrid" drainage system comprising conventional drainage and WSUD elements.

The natural, pre-development state of the catchment was used as the benchmark for the two systems. The development stage of the catchment is defined as that which occurs after a residential development has occurred, including all residential allotments, the road network and drainage infrastructure.

The common practice to design a conventional drainage system in Australia is to develop a DRAINS model for the development [39]. According to local government of South Australia guidelines, storm events at 10% annual exceedance probability (AEP) are considered as minor storms for designing conventional stormwater drainage systems in the DRAINS model, and those at 5% AEP are considered as major storms for checking the design. The 2016 design rainfalls provide more accurate estimates than Australian Rainfall and Runoff (ARR1987), combining contemporary statistical analysis and techniques with an expanded rainfall database [39]. Therefore, the 2016 Intensity–Frequency–Duration (IFD) design rainfalls with the 2016 edition of ARR2016 have been used for the design [39,40].

Hybrid drainage system refers to the developed catchment with a combination of conventional drainage system and WSUD elements, where stormwater comes from the roof of each allotment, and is managed by infiltration or reuse systems. The runoff that comes from roads and other pervious areas of total study area, excluding the roof area of each allotment, are managed by the conventional part of the hybrid system.

The hybrid drainage system is a multi-beneficial drainage system, because with WSUD, part of the stormwater can be saved for re-use, and the rest can be managed by a conventional drainage system.

The main aim of the paper was to compare the costs of the two drainage systems referred to above as conventional and hybrid systems for the modern 56-lot residential development at Pipers Crest in South Australia. These tasks are summarized as follows:

- Design of conventional and hybrid drainage systems for the selected catchment, using the DRAINS modeling tool and source control principles.
- Estimation of the installation cost of the two drainage systems based on the guidelines from the Australian literature, and
- Comparison of the performance and the cost-effectiveness of the two drainage systems by taking the pre-developed scenario as the benchmark.

## 2. Materials and Methods

### 2.1. Study Area

Pipers Crest is a residential subdivision located on the Fleurieu Peninsula of South Australia. The site is located 60 km southeast from the state capital, Adelaide, and sits on the banks of the Angas River [41]. The subdivision area is 6.71 ha, in which a total of 56 allotments were constructed between 2015 to 2019. The allotment areas vary from 491 $m^2$ (allotment no. 121 with a roof area of 198 $m^2$) to 2787 $m^2$ (allotment no. 146 with a roof area of 760 $m^2$), with a mean of 917 $m^2$. Roof area in the development ranges between 130 and 826 $m^2$ with a mean of 316 $m^2$. The mean annual rainfall and evapotranspiration are 491.9 and 1472.2 mm respectively, based on data 0.4 km and 2.6 km from the site (Australian Bureau of Meteorology Station 023747 and 024580) [42]. The site is generally dry in summer and wet in winter. The dominant soil type is considered in the study to be sandy clay. Based upon procedures in local stormwater management design guidelines—Australian Rainfall and Runoff—rainfall intensities for one-hour storm duration (design) at 10% and 5% AEPs are 23.7 and 28.3 mm/h, respectively [39]. The layout of the existing conventional drainage system at Pipers Crest was made available by the design consultant, Aurecon Australasia Pty Ltd. (An engineering and infrastructure consulting company), and consists of 42 on-grade single pits and 43 concrete pipes of varying sizes to convey stormwater to the adjacent creek. Based on this information, the detailed road and drainage networks are shown in Figure 1.

### 2.2. Study Approach

Two types of drainage systems, namely a conventional and a hybrid drainage system, were designed for 10% storm events, and were checked for 5% AEP storm events with a one-hour storm duration. In designing any conventional system, total areas including allotment (pervious and impervious) and road areas are considered as the contributing area; while in designing hybrid drainage systems, the roof area (only) of each allotment is managed by the WSUD elements, and the remainder is considered as contributing to the street drainage system. After designing both drainage systems, the installation cost of each system was estimated using local construction and WSUD cost estimation data [44,45]. The design and cost estimation procedure of both drainage systems are described in Sections 2.2.1–2.2.3.

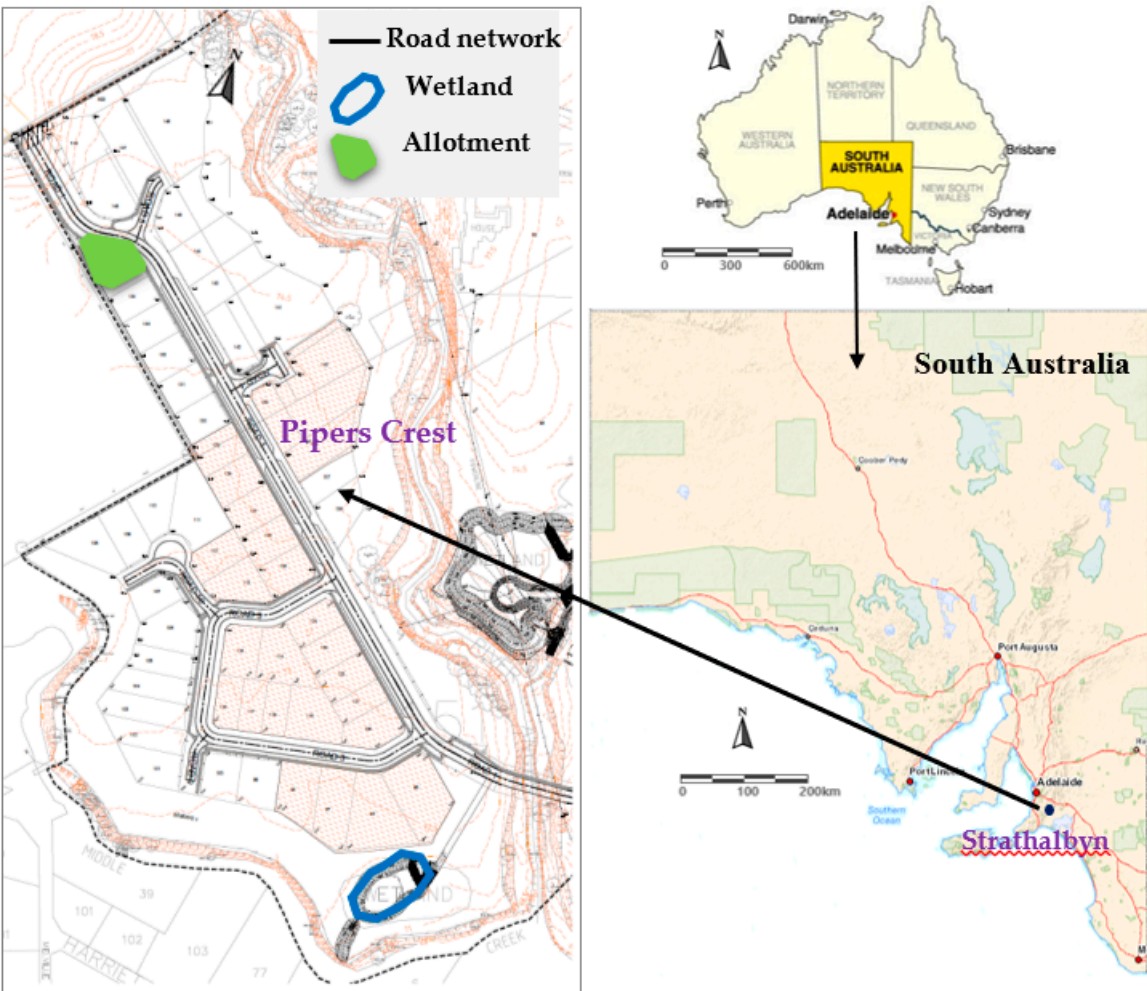

**Figure 1.** Pipers Crest development site, Strathalbyn, South Australia [43].

2.2.1. Design Procedure of Conventional Drainage System

DRAINS is a stormwater drainage system design and analysis tool that was first released in Australia in January 1998, and is widely used for urban stormwater system design and analysis in Australia and New Zealand at small to large scales [46,47]. DRAINS consists of several components, including ILSAX, the rational method, extended rational method (ERM) and ARR 2016 IL-CL for converting rainfall inputs to runoff; it also includes several flow routing procedures [48,49]. The ILSAX component of DRAINS is a relatively simple mathematical, event-based model, commonly used in Australia to simulate runoff flows in urban catchments based on the time-area routing procedure [49,50].

ILSAX uses Manning's equation for pipe routing and the Horton equation to estimate the infiltration losses of pervious areas. The DRAINS model needs design rainfall data, user specified pit details, sub-catchment details and travel time data, pipe layout and overflow route details as inputs. The "AS/NZS 3500.3.2018: Stormwater Drainage" provides specifications for selecting these elements of stormwater drainage systems. Finally, resulting runoff hydrographs and suitable pipe sizes are given as outputs. Though DRAINS is widely used for commercial and consulting work, only a few studies have adopted it for research [51,52]. We adopted the DRAINS stormwater drainage system design tool for identifying the appropriate pipe requirements for the case study site. The elements of the design procedure are described in the following sections.

The Hydrological Model Set Up

The ILSAX hydrological model requires the following to be nominated as input parameters: soil type and depression storage for impervious, supplementary and pervious areas. These are sensitive to both runoff volume and peak flow rate [53]. Depression storage is an initial loss: it is the depth of rainfall that is retained in depressions after infiltration on both pervious and impervious surfaces [54,55]. According to the DRAINS user manual, depression storage varies from 0 to 5 mm for impervious and supplementary areas and from 2 to 10 mm for pervious areas. The parameter "soil type "defines the infiltration process of pervious areas: it follows the US soil conservation service system of four soil types [56]. These are sand, sandy clay, medium clay and heavy clay soil types, respectively. In this paper, the values of depression storage were set as 1 mm for impervious and supplementary areas, and 5 mm for pervious areas, the latter value corresponding to that applicable to sandy clay soil type.

Rainfall Data Set Up

Any study involving design rainfall needs historical rainfall data or rainfall Intensity–Frequency–Duration (IFD) relationships [57,58]. In Australia, conventional drainage systems are generally designed using IFD relationships available from the Australian Bureau of Meteorology. IFD is defined as a three-way relationship to explain the statistical distribution of rainfalls occurring over different time periods in a study location [47,59,60]. In this study, the 2016 IFD design rainfall data were used, as they provide more up-to-date estimates. DRAINS needs values for nine parameters in order to calculate the IFD relationship at a specific location. These parameter values are: rainfall intensities at 50% and 2% AEPs for 1-, 12- and 72h durations; skewness, G of the log-transformed daily rainfall; frequency factors, F50 and F2 for 50% and 2% AEPs.

The study area, Pipers Crest, is located in Zone 6: the South Australian Gulf [61]. The critical storm duration was calculated to be one hour, based on the overland flow travel time estimation technique proposed by Argue [62]. The average rainfall intensities for the 10% and 5% AEPs events were calculated as 24.6 and 30.7 mm/h, respectively.

Pit Locations and Types

The pit in a storm drainage system is a node where water enters the drainage system. At a node point, the pipe can change its direction, slope or size [47]. A stormwater pit serves several purposes from a modeling perspective [63]. A node can be used to represent a street gully pit (or inlet), a manhole, a junction and a flow diversion [64,65]. There are two types of gully pits used in Australia: (a) on-grade pits and (b) sag pits. On-grade pits are generally located on slopes, and these allow bypass flows to travel past one gully pit to the next. Sag pits are in hollows or depressions, so that water cannot readily escape. The pressure (head) loss coefficient, $K_u$ is a dimensionless coefficient for full pipe flow, whose value varies with the geometry of the pit [66]. It defines the changes between the hydraulic grade line (HGL) and the total energy line (TEL) at a pit due to turbulence and other effects. The HGL is the most crucial property of a drainage system because it indicates the pressure level operating in a pipe or channel. The pressure loss coefficient, $K_u$ enables changes of HGL in a pipe system to be calculated.

The DRAINS manual suggests the value of $K_u$ to be 4 or 5 at the starting point (top point) of a drainage line, and 0.5, a conservative value, for pits in straight flow. However, $K_u$ of 1.5 is recommended when any pit changes the direction by more than 60°.

The DRAINS model includes a gully pit blockage factor taking account of the capacity reduction due to blockage by debris: it varies from 0 (fully open) to 1 (entirely blocked) [67,68]. Based on recommendations given in the DRAINS software manual, the blockage factor was assumed to be 0.2 for on-grade gully pits and 0.5 for sag pits.

As noted earlier, the Pipers Crest development consists of a total of 42 pits. Their locations were verified through field visits. On-grade gully pits were considered to have 3% crossfall and 1%

longitudinal grade. Finally, the surface elevation of each pit was determined using a contour map of the site.

Sub-Catchment Data

The input parameters of a DRAINS sub-catchment include: size, impervious (paved) area, supplementary paved area, pervious (grassed) area and travel time (or time of concentration) for runoff to concentrate at the catchment outlet [69]. The impervious area (road area and roof area), supplementary area (other paved, e.g., footpath and driveway) and pervious area of each sub-catchment were estimated from the Google map of Pipers Crest. For the conventional drainage system, the total impervious area included allotment roofs, driveways and road area. For the hybrid drainage system, only the road area fronting each allotment was considered to calculate impervious area, as it was assumed that allotment roof area was connected to an adequate infiltration system using the design procedures described below.

Travel Time Calculation

"Travel time" is a hydrological concept that provides the critical storm duration, and hence, critical flow rates in a catchment. It is the time required for stormwater to flow from the most remote point in a catchment to the outlet through the natural and formal flow paths. It is a function of the topography, geology and land use within a catchment [70]. In the DRAINS tool, separate travel time is required for pervious and impervious areas. The travel time was calculated using the procedures described by Argue [62]. As recommended by Argue [62], a 5 min travel time should be used for residential roofs where water is directly conveyed to the nearest drainage channel. Thus, a minimum travel time, 5 min, was adopted for the impervious area of each sub-catchment. The estimated travel time for the pervious area of the 42 sub-catchments varied from 10 to 60 mins.

Pipe Type and Details

A pipe is a circular, closed conduit that drains stormwater from one pit to another [71]. For pipe design in DRAINS, this includes variables relating to pipe type (material and class), the minimum pipe cover, Manning's roughness, the pipe length, pipe size, invert levels at the upstream and downstream ends and the pipe slope (%). Manning's roughness (n) value is the most commonly used parameter to account for losses in open channel flows. The flow velocity and flow rate at a cross section is calculated using Manning's equation. The total energy line (TEL) of each pipe defines the surface elevation of the connecting pit, while the pipe invert levels are the lowest point on its interior surface [48]. Upstream and downstream invert levels of each pipe are calculated by subtracting minimum pipe cover and pipe size from the total energy line level. The slope of each pipe was determined from upstream and downstream invert levels data and pipe length.

According to the "AS/NZS 3500.3.2018: Stormwater Drainage" and local council's (City of Alexandrina) guidelines, minimum values for pipe size, pipe cover and slope and maximum pipe length were adopted as 300 mm, 600 mm, 0.5% and 100 m, respectively. The material and class of pipe were selected as concrete, under roads: 0.5% minimum slope was adopted for each pipe in the study area. Pipe roughness (n) was entered as a default value for concrete, 0.013 s/m$^{1/3}$.

Overflow Route Details

Overflows are defined as the stormwater flows that bypass the conventional drainage system during events which exceed the design capacity [72]. Upwelling can happen when a downstream pipe has insufficient flow capacity. Overflow routes are used to define the hydraulic characteristics of the surface flow paths along which water flows from pits, headwalls or detention basins [47]. In the DRAINS model, the required inputs for overflow pathways are: the reach length of the channel, the overflow travel time between pits, the invert levels, the channel cross section, slope, safe depth and acceptable velocity. The cross section of the overflow route wizard produces a shape of cross section

that should be specified as physically realistic for the flows that will occur. The DRAINS tool selects pipe sizes in a way that ensures the resulting overflow from the design event meets the requirements of safe depths, velocity and flow rates. "Percentages of downstream areas" are used to define the quantity of sub-catchment flow appearing at the downstream pit in an overflow situation. This allows for flow characteristics to be calculated at all points along the overflow route.

Overflows follow the drainage pipe pathways, and hence the corresponding overflow reach lengths are identical to the underlying pipe lengths. According to the DRAINS manual, 2 mins of travel time is adopted for each overflow route. The surface elevations of the upstream and downstream pits were taken as the invert levels at the upstream and downstream ends of each overflow route, respectively. Based on field observations, the shape of each overflow route was decided to be a 7.5% roadway with 3% crossfall and barrier curb. According to the local council's guidelines (those of the Alexandrina city council) for stormwater system design, the safe depth for major storm and minor storm and the safe depth ×velocity values were set as 0.3 m, 0.15 m and 0.4 m$^2$/s, respectively [73].

Detention Basin

Stormwater management design guidelines for local councils [73,74] require several design criteria. One of the most important is that the outflow peak flow at the post-development stage should not exceed the pre-development outflow rate for both minor and major storm events. To meet this guideline, a downstream detention basin is typically included to limit and control post development flows to pre-development flows in urban developments. The input characteristics that are required for designing a detention basin in DRAINS include: initial water level, storage infiltration characteristics (elevation–perimeter and hydraulic conductivity), an elevation–surface area (or elevation–volume) relationship and a low-level outlet specification. The low-level outlet includes five options: an orifice acting as a free outfall, a pit or sump outlet, a circular conduit, a rectangular channel and no outlet.

An orifice outlet was selected as the low-level outlet in this project. Based on guidelines provided by the local government authority [74], the detention basin was assumed to be a trapezoidal prism with an average coefficient of discharge ($C_d$) of 0.61, 1:5 ratio batters and a maximum water depth of 1.2 m. The diameter of the orifice plate was taken as 140 mm and 160 mm for 10% and 5% AEP events, respectively, where the pipe diameter was 375 mm; the center elevation was taken as 68.30 m. $C_d$ depends on the ratio of orifice to pipe diameter called the beta ratio and Reynold's number [75]. The beta ratio was calculated as 0.37, and $C_d$ has been found as 0.61 from the following Figure 2.

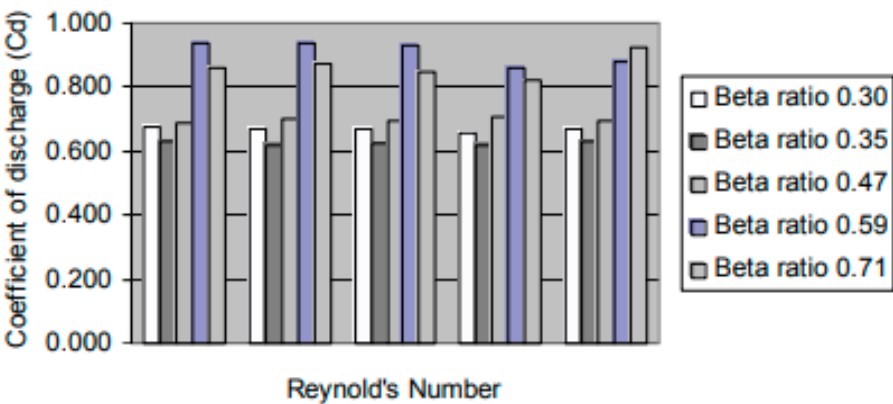

**Figure 2.** Bar diagram of coefficient of discharge ($C_d$) for different beta ratios

Inputs into the infiltration data wizard were; perimeter and elevation of the two detention basin cases and hydraulic conductivity of sandy clay soil ($3 \times 10^{-5}$ m/s). These data enabled the infiltration losses from basin areas to be calculated. Detailed design of the two detention basins is presented in Section 3.2.

Dummy Outlet

The outlet node is the end or the receiving point of a drainage system: it can be connected to a detention basin [47]. The dummy outlet was placed at the end of the last pipe (Pipe43), as this particular pipe cannot re-enter the system due to the surface elevation constraints of the site.

Design Criteria

After defining all necessary design inputs, the DRAINS model was run many times under conditions of the selected storm events (minor and major), varying pipe sizes until no flooding, adequate freeboard at all pits and 'safe' flows occurred through all overflow routes. The detailed results of the model output are presented in Section 3.

### 2.2.2. Design Procedure of Hybrid Drainage System

The design of hybrid WSUD systems has been previously described by Akhter et al. [76]. In the project at Pipers Crest, hydraulic designs of three infiltration systems (a leaky well, soakaway or infiltration trench for each allotment) were undertaken simultaneously for the sandy clay soil condition using procedures developed by Argue [22]. As in conventional system design, the hybrid systems were also designed for 10% and 5% AEP storm events. One of the three infiltration systems was then selected based on satisfactory performance in meeting clearance distance criteria for footings and property boundaries (2 m for sandy clay soil), as well as emptying time criteria (2 days for 10% AEP and 2.5 days for 5% AEP storm events). In any allotment where all three infiltration systems failed to meet the design criteria, the option of using a large rainwater tank was available. Finally, from the possible WSUD systems (infiltration systems/rainwater tank), the most economical one was selected to install in each allotment. The detailed design outcomes of the infiltration systems investigation in the sandy clay soil condition is summarized in Section 3.1.

### 2.2.3. Installation Cost of Drainage Systems

As explained above, this study sets out to design two drainage systems, one conventional, the other hybrid, and to compare the costs of installation of each. Taking account of the two flood conditions under consideration (minor and major events), this leads to a total of four scenarios to be costed. Installation cost of any system consists of two major components; material cost and labor cost.

In this costing process, two sources were used: Australian cost estimation dataset [45] and local construction cost estimation data [44]. In costing a conventional drainage system, the following components are to be considered:

- trench excavation,
- backfilling trenches using the excavated material,
- pipes
- dual rubber ring joints for pipes,
- precast concrete base, walls and pipe cover,
- gully pits,
- inclined bend from one pit to another, and
- detention basin construction cost.

The cost of the detention basin includes trench excavation (exceeding 300 mm wide) by machine with 1000/2000 mm total depth and extra cost for disposal of surplus excavated material in sandy clay soil.

On the other hand, the cost components of a hybrid drainage system include those of the supporting conventional system together with the following components of the infiltration systems:

- excavation for the infiltration system,

- installation of the geofabric liner,
- placement of a perforated pipe, gravel storage layer and topsoil layer,
- application of grass seed, fertilizer and watering, and
- the rainwater tank unit, where applicable.

The items that are to be considered in estimating the cost of a rainwater tank unit include the supply and installation of a round, galvanized tank, pumps, plumbing and fittings, float system and concrete base.

The cost of pipes, pits and the detention basin was estimated in Australian dollars ($AUD), using Rawlinsons' unit rates in 2018 [45]. The cost of WSUD systems was estimated in $AUD using the WSUD technical manual 2010 [44] and was converted into 2018 using an inflation calculator [77]. The details of the costing process are presented in Section 3.6.

## 3. Design Outcomes of the Two Drainage Systems

### 3.1. Design Result of WSUD Systems

In the paper published by Akhter et al. [76], it was found that satisfactory performance for draining the roof areas of all allotments located in sandy clay should be achieved using the available three infiltration systems (leaky well, soakaway or infiltration trench) in combinations with a 1 KL rainwater tank. However, when costs of these alternatives were considered, the leaky well emerged as the preferred option. In particular, considering the design outcomes of 10% AEP events, it was found that installation of one leaky well was required for 54 allotments, while two leaky wells were required for two large allotments (no. 146 with roof area 826 m$^2$ and no. 140 with roof area 769 m$^2$). When the consideration was 5% AEP events, leaky well was also found to be the most economical option. However, two leaky wells were needed for an additional allotment (no. 135 with roof area 497 m$^2$).

### 3.2. Detention Basin Design: Dimensions, Inflow Rate, Outflow Rate and Water Levels

A detention basin fitted with an orifice plate outlet was placed at the end of both the conventional and hybrid drainage systems to limit discharge in accordance with the local government requirement that the outflow peak after development should not exceed that from the pre-development site. The pipe40 (diameter 675 mm, length 7.55 m and slope 1.32%) which drains directly to the detention basin, was found to be the pipe with the highest peak flow rate.

As a result, pipe40 needed to be the largest pipe to effectively deliver stormwater to the detention basin of the conventional drainage system. It was required that the detention basin should be able to attenuate pipe40 peak flow to the pre-development peak flow rates of 0.045 m$^3$/s and 0.060 m$^3$/s for 10% and 5% AEP events, respectively.

The dimensions of the detention basins for the conventional and hybrid drainage systems are shown in Figure 3a–d, respectively.

These results show that the required detention basin volumes for conventional and hybrid drainage systems are 1785.60 m$^3$ and 738 m$^3$, respectively, under 10% AEP events. Similarly, in 5% AEP events, the required detention basin volumes for conventional and hybrid drainage systems are 2239 m$^3$ and 943 m$^3$, respectively. These outcomes show that the detention volume requirement to maintain the pre-developed peak runoff can be reduced by more than 50% in the hybrid drainage system, compared with the requirements for the conventional system, for both minor and major storm events. Details of the detention basin design for 10% AEP events are presented in Appendix A.

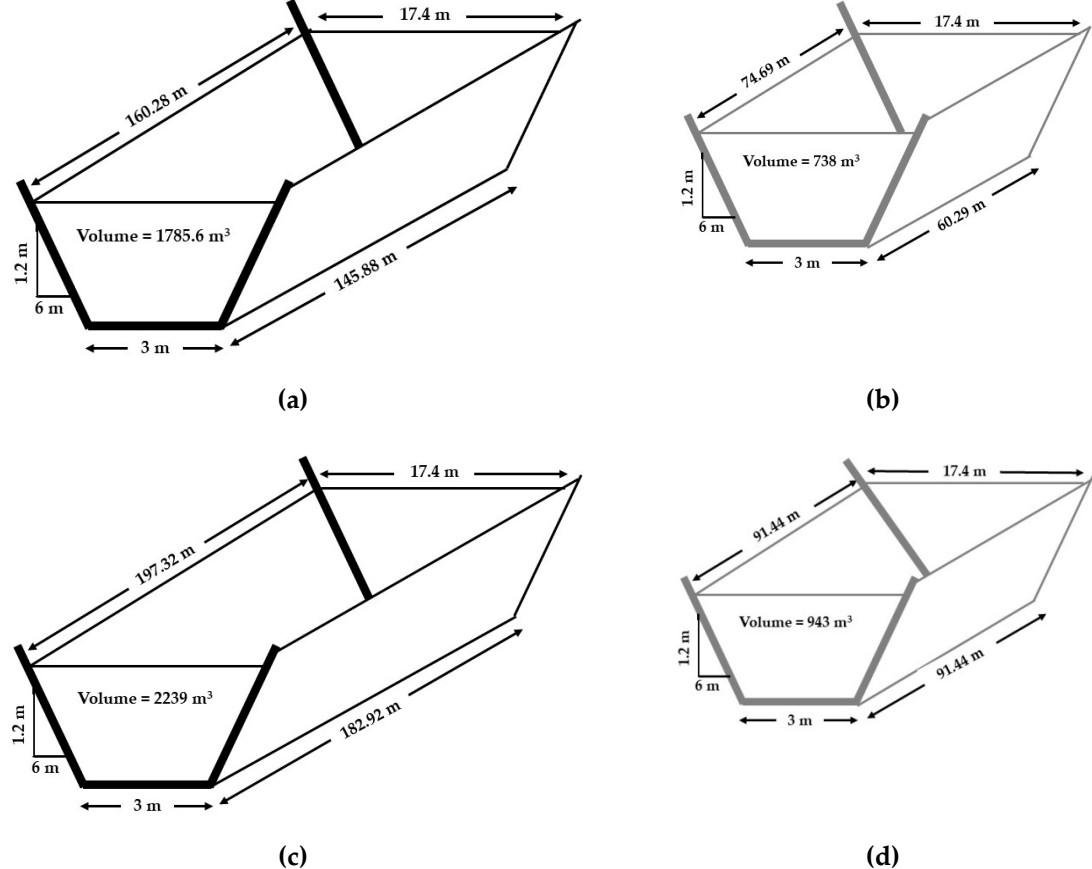

**Figure 3.** Detention basin dimension for minor storm (**a**) conventional and (**b**) hybrid drainage system and major storm (**c**) conventional and (**d**) hybrid drainage system.

The question: how effective are the detention basins likely to be meeting the council requirements relating to flow attenuation. The question is answered in Table 1, following.

**Table 1.** Comparison of design output of detention basin for the two drainage systems.

| Design Aspect | Minor Storm | | Major Storm | |
|---|---|---|---|---|
| | Conventional Drainage System | Hybrid Drainage System | Conventional Drainage System | Hybrid Drainage System |
| Maximum outflow rate (m³/s) | 0.034 | 0.032 | 0.048 | 0.045 |
| Maximum water level (m) | 69.16 | 69.17 | 69.28 | 69.29 |
| Storage volume (m³) | 667.11 | 269.81 | 867.78 | 365.78 |

It can be seen in Table 1 that the maximum outflow rates from the detention basin at 10% AEP are 0.034 and 0.032 m³/s, respectively (for the two systems), which are significantly smaller than the pre-development peak flow of 0.045 m³/s. While the peak values at 5% AEP are 0.048 and 0.045 m³/s (for the two systems), which are also much smaller than the pre-development peak flow of 0.060 m³/s.

The explanation for this is linked to an additional council requirement that water depth in the detention basin be limited to 1.2 m. The lowest design level of the basin was 68.33 m. Hence, the maximum allowable water level should be 69.53 m. Table 1 shows variation of the water level at 10% and 5% AEP for the two drainage systems. Taking account of this consideration the maximum water levels achieved at 10% AEP are 69.16 and 69.17 m, respectively, for the conventional and hybrid drainage systems. The values at 5% AEP are 69.28 and 69.29 m. Consequently, the maximum water level in each scenario is below the maximum allowable level of 69.53 m. Therefore, the detention system satisfies the council requirements and freeboard does not exceed maximum depth for each case.

Table 1 illustrates the variation of storage volume during minor and major storm events for the two drainage systems. When the conventional drainage system is considered, the maximum storage volumes of the detention basin for 10% and 5% AEP storms are 667.11 and 867.78 m³, respectively: values for the hybrid system are 269.81 and 365.78 m³. These values are well below the full capacity of 1785.60, 2239, 738 and 943 m³, respectively. So, the maximum storage volume for both drainage systems does not exceed the design volume of the detention basin during minor and major storms.

### 3.3. Vulnerability of Flooding

As noted earlier, the hydraulic grade line (HGL) indicates water level variation along the pipes. HGL at the pit locations indicates if there is a possibility of upwelling. Upwelling will happen when HGL is at the total energy line (TEL). One of the design requirements is to have enough height difference (freeboard) between TEL and HGL for all 42 pits, so that the safety of the public is assured. TEL and HGL were recorded at each pit from the DRAINS output. Freeboard was produced by analyzing the height difference between HGL and TEL, using Microsoft Excel for the two drainage systems. Large values of freeboard at a pit imply less chance of upwelling (in the pit) during flood events; zero overflows may be possible under this scenario. Small values of freeboard imply the greater chance of pit overflows. So, large freeboard is better to get later or no water upwelling from any pit to make safe the study site from the risk of flood. Figure 4a,b produced by analyzing the freeboard values of 42 pits resulted from the consideration of minor and major storms. It is clear from Figure 4 that median and minimum freeboard values of the hybrid system are greater than those from the conventional system. This indicates that the hybrid system gives a better safety margin than the conventional system in both minor and major storms.

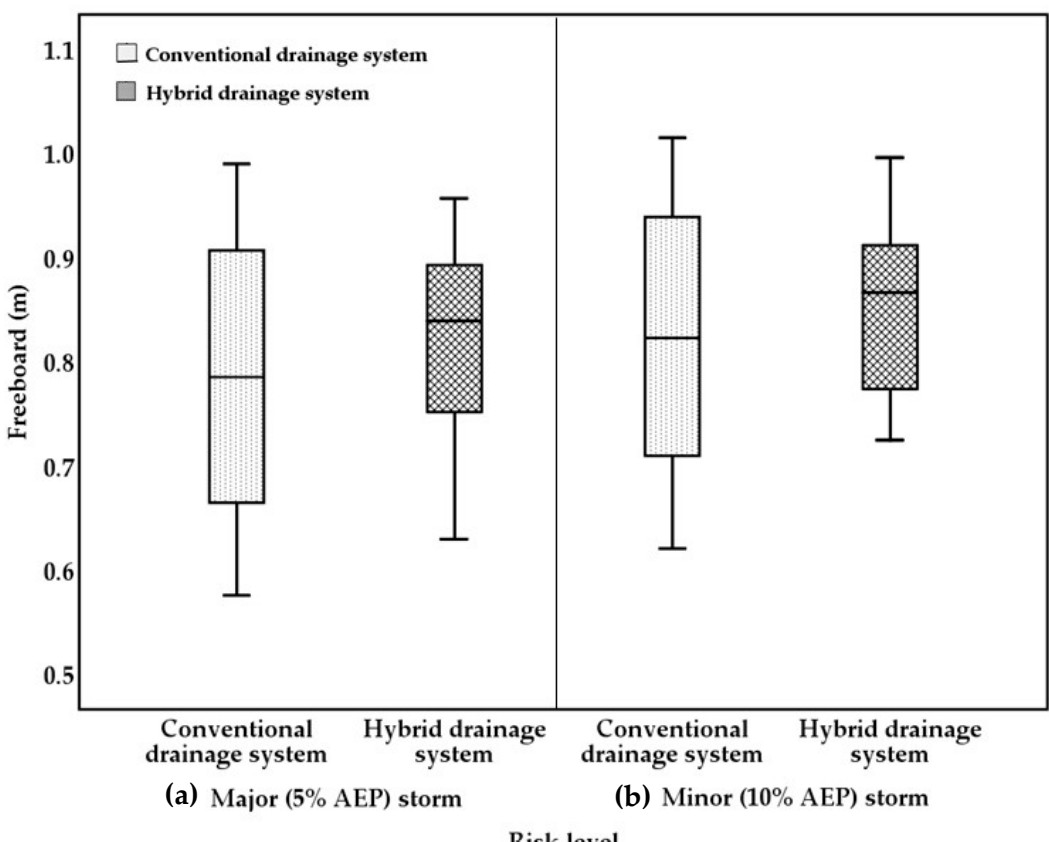

**Figure 4.** Freeboard variation determined for conventional and hybrid drainage systems for (**a**) 5% and (**b**) 10% annual exceedance probability (AEP) storm events.

The freeboard data were statistically analyzed to assess if the observed differences in Figure 4 are significant. Table 2 shows the significance results of the normality test.

**Table 2.** Normality test of freeboard for the two drainage systems.

| Normality Test | Minor Storm | | Major Storm | |
|---|---|---|---|---|
| | Conventional Drainage System | Hybrid Drainage System | Conventional Drainage System | Hybrid Drainage System |
| Kolomogorov–Smirnov | 0.191 | 0.000 | 0.028 | 0.200 |
| Shapiro–Wilk | 0.011 | 0.000 | 0.000 | 0.240 |

Table 2 shows that the freeboard conditions for the hybrid system when subjected to minor events, and that from the conventional system when subjected to major events, are not normally distributed (Kolomogorov–Smirnov test). Therefore, the significance between the median freeboard of the two systems (at minor and major events) was separately assessed through the non-parametric method (Wilcoxon Signed Rank test). Table 3 (below) shows that the freeboard values of the two systems are significantly different only at major events. Therefore, the hybrid system shows significantly reduced risk of flooding (upwelling at the pits) at major events.

**Table 3.** Hypothesis test of freeboard for the two drainage systems.

| Null Hypothesis Test | Significance Value | Storm Events | Decision |
|---|---|---|---|
| Median freeboard of the two systems | 0.117 | Minor storm | Values are not significantly different ($>0.05$) |
| | 0.000 | Major storm | Values are significantly different ($<0.05$) |

### 3.4. Comparison of Pipe Size and Length

As noted in Section 2, there were a total 43 pipes and 42 pits in the study site. The pipe sizes and lengths were adjusted in the DRAINS model according to local government requirements and commercially available pipe sizes. Pipe sizes varied from 300 mm to 675 mm in the two drainage systems. The pipe sizes, total lengths and the percentages of pipe lengths are summarized in Table 4. The results indicate that a larger proportion of smaller diameter pipe were required for the hybrid system compared to the conventional system. For example, pipes of 675 mm diameter are required to cover 248.59 m of the conventional drainage system, while the maximum pipe diameter needed for the hybrid system was 600 mm for a shorter length of 101.56 m.

**Table 4.** Comparison of pipe size and length between two drainage systems.

| Pipe Size (mm) | Conventional Drainage System | | Hybrid Drainage System | |
|---|---|---|---|---|
| | Total Length of Pipes (m) | Percentage of Pipe Length | Total Length of Pipes (m) | Percentage of Pipe Length |
| 300 | 207.01 | 20% | 313.53 | 30% |
| 375 | 162.98 | 15% | 72.38 | 7% |
| 450 | 24.96 | 2% | 126.67 | 12% |
| 525 | 235.72 | 22% | 442.56 | 41% |
| 600 | 177.44 | 17% | 101.56 | 10% |
| 675 | 248.59 | 24% | | |
| Total | 1057 | 100% | 1057 | 100% |

Overall, smaller pipe sizes are required for the hybrid drainage system compared to those required for the conventional drainage system. The reason for this is that an effective impervious area and hence the resulting runoff volume in the hybrid drainage system was smaller than that in the conventional drainage system because only roof runoff was managed by the WSUD elements.

## 3.5. Comparison of Overflow Routes

Overflow routes are commonly the flow paths along road surfaces. The overflow route where the highest flow rate is observed for minor and major events can be seen in the hydrograph plots in Figure 5.

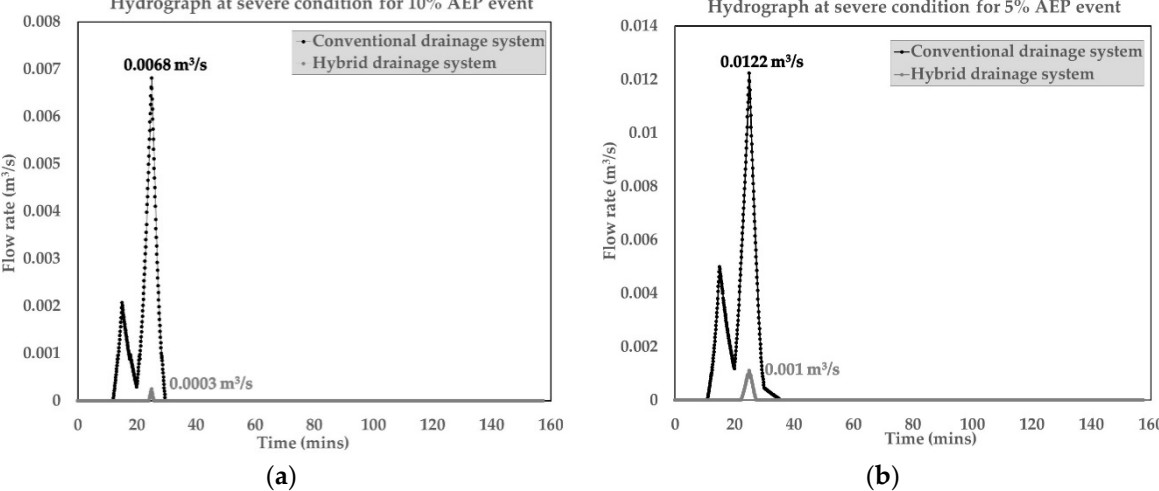

**Figure 5.** Comparison of overflow routes for (**a**) 10% and (**b**) 5% AEP storm events.

Peak overflows resulting from both minor and major storms in the hybrid system are almost zero. However, with the conventional system, peak overflows varied from 6.8 L/s to 12.2 L/s for 10% and 5% AEP events.

## 3.6. Installation Cost of Drainage Systems

The installation costs of the conventional and the hybrid systems were estimated based on the cost components given in Section 2.2.3. The installation costs of the detention basin, pipes, pits and other items determined in this study are presented in Tables 5–7. The cost of the conventional drainage system is quantified by the total installation cost of each item in $AUD using the unit rates in 2018 [45].

The installation costs of various items of (on-site) WSUD systems for the smallest allotment of Pipers Crest was calculated in $AUD using the WSUD technical manual 2010 [44] and are presented in Table 7. Then the cost was converted into 2018 values using inflation calculator [77].

The total installation cost of the conventional drainage system has been estimated as $603.17k and 637.63k designed for 10% and 5% AEP events, respectively. However, for the hybrid drainage system, the total installation costs have been estimated as $494.66k and $521.17k for 10% and 5% AEP events, respectively. The installation cost of the hybrid drainage system is about 18% less than that of the conventional drainage system for both 10% and 5% AEP storm events.

A major factor in achieving a smaller cost for the hybrid drainage system compared to the conventional system is the significantly smaller detention basin volume required.

**Table 5.** Cost of detention basin for both drainage systems.

| Item | Price, $AUD | Unit | Storm Event | Drainage System | Detention Basin Size, m³ | Total Cost, $AUD |
|---|---|---|---|---|---|---|
| Exceeding 300 mm wide trench excavation by machine with 1000/2000 mm total depth | 76 | $/m³ | Minor | Conventional | 1785.60 | 135,706 |
| | | | | Hybrid | 738 | 56,088 |
| | | | Major | Conventional | 2239 | 170,164 |
| | | | | Hybrid | 943 | 71,668 |

**Table 6.** Cost of different items of the conventional drainage system (pipe and pit network).

| Item | Price, $AUD | Unit | Pipe Size, mm | Total Pipe Length, m or Total Pit no. | Total Cost, $AUD |
|---|---|---|---|---|---|
| **Pipe Details** | | | | | |
| 300 mm wide trench excavation | 11.75 | $/m | 300 | 207.01 | 2432 |
| Exceeding 300 mm wide trench excavation | 77.5 | $/m | | 581.08 | 45,034 |
| 500 mm soil depth backfilling with excavated material | 65 | $/m³ | 300 | 7.73 | 75 |
| 1000 mm soil depth backfilling with excavated material | 65 | $/m³ | 375 | 74 | 1804 |
| Dual rubber ring joints | 138 | $/m | 300 | 207.01 | 28,567 |
| **Pit Details** | | | | | |
| Precast base and walls with 900 mm diameter & 900 mm deep | 1005 | $/no. | | 42 | 42,210 |
| Extra cost for each additional 100 mm in depth | 72 | $/no. | | 42 | 3024 |
| Precast concrete cover | 160 | $/no. | | 42 | 6720 |
| **Pipe bend** | | | | | |
| 45 degree | 360 | $/no. | 300 | | 360 |
| | 610 | $/no. | 450 | | 610 |
| | 1035 | $/no. | 525 | | 1035 |
| 88 degree | 760 | $/no. | 375 | | 760 |

**Table 7.** Unit cost of smallest allotment (area 496 m²) for the water sensitive urban design (WSUD) process installation.

| Item | Quantity | Unit | Rate | Cost ($/m) |
|---|---|---|---|---|
| Excavate and stockpile | 2.93 | m³/m | 20 | 58.6 |
| Supply and install geofabric liner | 59 | m²/m | 5 | 295 |
| Supply and place perforated pipe (100 mm diameter) | 5 | m/m | 13 | 65 |
| Supply and place gravel storage layer | 2.93 | m³/m | 65 | 190.45 |
| Supply and place topsoil layer (300 mm minimum thick) | 0.44 | m³/m | 70 | 30.8 |
| Supply and apply grass seed, fertilizer and watering | 1.47 | m²/m | 1 | 1.47 |
| Indicative 1 KL rainwater tank system costs | | | | $1590 |

Note: Cost = Quantity * rate.

## 4. Conclusions

This paper has explored the drainage infrastructure comparison of two categories of drainage systems applied to a developed residential sub-division of a modern suburb in South Australia. The objectives were to design a conventional and a hybrid drainage system for Pipers Crest using the DRAINS modeling tool and for the installation of both systems. Pipe size, length, detention basin and other important elements were modeled for both drainage systems, taking account of the local council's guidelines and the DRAINS manual. The impervious, supplementary and pervious area percentages were properly classified and entered for each drainage system. DRAINS output showed adequate freeboard, safe flow, and no water upwelling from any pit for both 10% and 5% AEP events. The stormwater flow was passed through upstream pipes to downstream pipes and finally discharged through a detention basin at the outlet of the study catchment. Considering the installation cost, the hybrid drainage system was more economic, strongly influenced by the cost of the larger detention basin required by the conventional drainage system. The hybrid system has reduced material consumption and reduced urbanization impact downstream when compared to the conventional system

In this paper, the performances of both drainage systems have been performed for 10% and 5% AEP events. Checking the performance of the system when exposed to a much greater storm intensity such as the 1% AEP, was considered out of scope with this typical design scenario, but will

be considered for future work. The findings of this paper may contribute to the economic analysis of WSUD systems by other researches. The installation cost of the hybrid drainage system has been estimated at the development stage by adding the cost of WSUD components. However, WSUD systems have several additional direct and indirect benefits, which have not been examined in this paper. In future, life cycle cost and benefit of WSUD systems need to be estimated to assess the full economic feasibility of WSUD systems.

**Author Contributions:** Conceptualization, F.Ak., G.A.H., F.A., B.M. and J.R.A.; methodology, F.Ak., G.A.H., F.A. and B.M.; software, F.Ak., G.A.H., F.A. and B.M.; validation, F.Ak., G.A.H., F.A. and B.M.; formal analysis, F.Ak.; investigation, G.A.H., F.A., B.M. and J.R.A.; resources, G.A.H., F.A. and B.M.; data curation, F.Ak.; writing—original draft preparation, F.Ak.; writing—review and editing, G.A.H., F.A., B.M. and J.R.A.; visualization, F.Ak., G.A.H., F.A., B.M. and J.R.A.; supervision, G.A.H., F.A., B.M. and J.R.A.; project administration, G.A.H., F.A., B.M. and J.R.A. All authors have read and agreed to the published version of the manuscript.

**Funding:** This research is funded by the Australian Government Research Training Program Scholarship (RTPs) awarded to the primary author.

**Conflicts of Interest:** We declare no conflict of interest between the authors.

## Appendix A

*Appendix A.1. Detention Basin Design for Conventional Drainage System (Figure 2a,b)*

The critical storm duration was calculated as 60 min using urban stormwater runoff design guidelines (Argue 1986). For a 60 min storm duration, the rainfall intensity was found to be 24.6 mm/h from BOM 2018. Total area of the catchment was 67,100 m$^2$. The pre-development flow was calculated for a 10% AEP storm event as follows:

Runoff coefficient, $C_{10\,pervious}$ = 0.10; Conversion factor, $F_Y$ = 1.0; C = $C_{10\,pervious} \times F_Y$ = 0.10 × 1.0 = 0.10

$$Q_{pre-development} = \frac{CIA}{360} \tag{A1}$$

$$Q_{pre-development} = \frac{0.10 \times 24.6\;mm/h \times 67100\;m^2}{360 \times 10^4} = 0.045\;m^3/s$$

The DRAINS model simulated a post-development flow:

For a conventional drainage system, $Q_{Post-development\,(10\%\,AEP)}$ = 0.541 m$^3$/s

For the hybrid drainage system, $Q_{Post-development\,(10\%\,AEP)}$ = 0.250 m$^3$/s

To limit discharge to pre-development flows, an orifice plate is required before the drainage outlet. Discharge through a circular orifice plate,

$$Q = C_d A \sqrt{2gH}\;\left[\text{where } A = \frac{\pi}{4}D^2\right] \tag{A2}$$

Here, average coefficient of discharge ($C_d$) = 0.61 and a maximum water depth, H = 1.2m as per Council guidelines.

So, the diameter of orifice, $D = \sqrt{\frac{4Q}{\pi C_d \sqrt{2gH}}} = \sqrt{\frac{4 \times 0.045}{\pi \times 0.61 \sqrt{2 \times 9.81 \times 1.2}}}$ = 0.140 mm = 140 mm

Required detention basin volume size:

For conventional drainage system, V = ($Q_{Post-development} - Q_{Pre-development}$) × (Time)
= (0.541 − 0.045) × 60 × 60 = 1785.60 m$^3$

For hybrid drainage system, V = ($Q_{Post-development} - Q_{Pre-development}$) × (Time)
= (0.25 − 0.045) × 60 × 60 = 738 m$^3$

The basin design is based upon a trapezoid shape and the geometric properties are shown as follows:

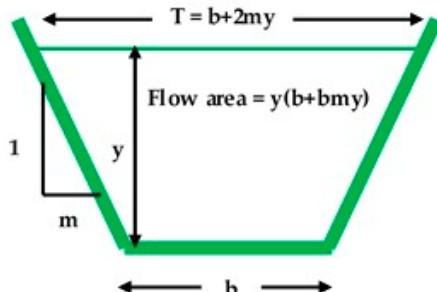

**Figure A1.** Design of Trapezoidal detention basin

**Table A1.** Design parameters of detention basin for two drainage systems.

| Conventional Drainage System | Hybrid Drainage System |
| --- | --- |
| Using geometric formulas, the following basin design parameters have been obtained: $b$ = 3 m, for 1V:5H batters $y$ = 1.2 m and $m$ = 6 m, Top Width $B$ = $b + 2my$ = 17.40 m, Flow Area = $y \times (b + bmy)$ = 12.24 m$^2$, Bottom length, $L_b$ = $V/A$ = 145.88 m. Top length, $L_t$ = $L_b + 2my$ = 160.28 m | Using geometric formulas, the following basin design parameters have been obtained: $b$ = 3 m, for 1V:5H batters $y$ = 1.2 m and $m$ = 6 m, Top Width $B$ = $b + 2my$ = 17.40 m, Flow Area = $y \times (b + bmy)$ = 12.24 m$^2$, Bottom length, $L_b$ = $V/A$ = 60.29 m. Top length, $L_t$ = $L_b + 2my$ = 74.69 m |

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
