# Peer review of "Performance Evaluation of Stormwater Management Systems and Its Impact on Development Costing"

_water, doi:10.3390/w12020375_

Round 1
Reviewer 1 Report
This paper considers the design of two different drainage options in a small catchment in Australia. Standard design procedures and modelling tools are applied in combination with a state-of-the-art approach for estimating installation cost.
While this work would make for a fine consultancy report, I unfortunately really don't see any novel contribution to the scientific literature and thus suggest rejecting the paper.
Author Response
Reviewer 1
This paper considers the design of two different drainage options in a small catchment in Australia. Standard design procedures and modelling tools are applied in combination with a state-of-the-art approach for estimating installation cost.
While this work would make for a fine consultancy report, I unfortunately really don't see any novel contribution to the scientific literature and thus suggest rejecting the paper.
Response 1: A vital element of this paper is to establish the credibility of the paper in the eyes of practitioners who are its main target. The paper’s “novel contribution” is its finding that by taking an unconventional design approach, practitioners can not only reap environment benefits in residential subdivisions but, also, save money.
Reviewer 2 Report
The adopted value of CD = 0.61 (line number: 267, 488) requires justification (e.g. hydraulic diagram).
Line number: 232 - The empirical coefficient (n), for the Manning formula, has the dimension: s/m1/3.
Author Response
Reviewer 2
The adopted value of CD= 0.61 (line number: 267, 488) requires justification (e.g. hydraulic diagram).CD value has been justified using a reference and the text has been amended to make this clearer in Lines 269 to 274.
“An orifice outlet was selected as the low-level outlet in this project. Based on guidelines provided by the local government authority [74], the detention basin was assumed to be a trapezoidal prism with an average coefficient of discharge (CD) of 0.61, 1:5 ratio batters and a maximum water depth of 1.2 m”
Line number: 232 - The empirical coefficient (n), for the Manning formula, has the dimension: s/m1/3.Thank you for the comment. The text has now been amended and the dimension has been provided in Line 232 as follows: Pipe roughness (n) was entered as default value for concrete, 0.013 s/m1/3.
Reviewer 3 Report
This paper presents a comparison of two urban drainage systems for an urban area and shows the benefits of WSUD in terms of flood reduction. This paper is well written and technically sound. It provides more evidence on the adoption of WSUD for urban stormwater management. The following comments should be addressed before it can be published.
Literature review: the impact of population growth and urbanisation is well documented in the literature and thus there is no need for a detailed review here. The first two paragraphs could be summarised in one or two sentences. I would suggest the introduction focuses on the review of the WSUD in more detail, including the conceptual development, design approaches and standards and the impacts of various components (measures). This is more relevant to the aim of this paper.
Why is freeboard used as a performance indicator for comparison? How is this indicator related to flooding? If flooding is concerned, why not to use flood volume and flood depth as an indicator directly, which is commonly used in the literature see Want et al., 2019? Also, more explanations should be provided regarding how the freeboard is calculated: at one point or multiple points?
Vulnerability of flooding is mentioned in the paper. But what is the vulnerability of flooding and how it can be measured? Is it a component of flood risk? More explanations should be provided, in particular about how it is linked to flood risk.
Figure 3 shows the freeboard values are very similar for the 5% and 10% AEP events. These two events are equivalent to return periods of 20 and 10 years. I wonder, why are the differences in freeboard so small? How different are these two rainfall events?
The systems are assessed using the two AEP events. I wonder, how the systems could cope with a more extreme event such as an AEP of 1%? Nowadays, it is important to assess the resilience of stormwater systems, see Wang et al., 2019. I would suggest the authors to discuss this aspect of the designed systems. Further, the economic aspect (costs) is only one dimension of the concept of sustainability, see Casal-Campos et al. (2018). It is suggested to discuss this in the context of sustainability including social benefits.
‘Sustainable Urban Drainage Systems (SUDS)’ should be written as ‘Sustainable Drainage Systems (SuDS)’
Wang et al., (2019) Assessing catchment scale flood resilience of urban areas using a grid cell based metric, Water Res, volume 163, DOI:10.1016/j.watres.2019.114852.
Casal-Campos et al., (2018) Reliable, Resilient and Sustainable Urban Drainage Systems: An Analysis of Robustness under Deep Uncertainty, Environmental Science and Technology, 52(16), 9008-9021, DOI:10.1021/acs.est.8b01193.
Author Response
Reviewer 3
This paper presents a comparison of two urban drainage systems for an urban area and shows the benefits of WSUD in terms of flood reduction. This paper is well written and technically sound. It provides more evidence on the adoption of WSUD for urban stormwater management. The following comments should be addressed before it can be published.
Literature review: the impact of population growth and urbanisation is well documented in the literature and thus there is no need for a detailed review here. The first two paragraphs could be summarised in one or two sentences. I would suggest the introduction focuses on the review of the WSUD in more detail, including the conceptual development, design approaches and standards and the impacts of various components (measures). This is more relevant to the aim of this paper.The focus of the current paper was on design of conventional and hybrid drainage systems at catchment scale using the DRAINS modeling tool and to compare the resulting development cost. This article presents only a part of the first author’s PhD research project. Preliminary work including a review which covered the conceptual development, design approaches and standards and the impacts of various components (WSUD elements) have been presented in a prior publication titled “Selection of appropriate water-sensitive systems for stormwater quantity control in South Australia” (Akhter et al., 2018). Hence without repeating the same material we have given reference to this article [Section 2.2.2. Design procedure of Hybrid drainage system and Section 3.1. Design result of WSUD systems]. Noting the need for some background to WSUD, we have modified the text and updated the reference number in the manuscript (lines 53-62 of Section 1.1) to make our intention clearer.
Why is freeboard used as a performance indicator for comparison? How is this indicator related to flooding? If flooding is concerned, why not to use flood volume and flood depth as an indicator directly, which is commonly used in the literature see Want et al., 2019? Also, more explanations should be provided regarding how the freeboard is calculated: at one point or multiple points?When the hydraulic grade line (HGL) is at surface level (TEL), water is upwelled, indicating that there is zero freeboard and flooding will occur. Therefore, freeboard is considered as the most suitable indicator to quantify vulnerability of a location/point for flooding. A pit with more freeboard has a bigger safety margin for flooding than one with smaller freeboard. Therefore, freeboard is an appropriate measure to compare performance of the two systems.
In this article, one of the design requirements was to have enough height difference (freeboard) between TEL and HGL so that the occurrence of surface flooding is minimised during storms of particular intensity and duration. Some modification has w been added with previous explanation in the revised manuscript, now in Lines 393-399 of section 3.4.
Vulnerability of flooding is mentioned in the paper. But what is the vulnerability of flooding and how it can be measured? Is it a component of flood risk? More explanations should be provided, in particular about how it is linked to flood risk.The measurement of flood vulnerability was based on the determination of freeboard. This is detailed in the responses above to Reviewer 3, Question 2.
Figure 3 shows the freeboard values are very similar for the 5% and 10% AEP events. These two events are equivalent to return periods of 20 and 10 years. I wonder, why are the differences in freeboard so small? How different are these two rainfall events?Freeboard was calculated at 42 pits for both rainfall events based on model output. In this figure it shows that freeboard varies from 0.576 to 0.99 m for the conventional and 0.63 to 0.957 m for the hybrid drainage system when exposed to 5% AEP events. Whereas, under 10% AEP events, freeboard varies from 0.621 to 1.015 m for conventional and 0.725 to 0.996 m for the hybrid drainage system. The rainfall intensity was estimated as 30.7 and 24.6 mm/hr of a 1 hour critical storm for 5% AEP and 10% AEP events at this location, respectively. According to the “AS/NZS 3500.3.2018: Stormwater Drainage” and local council’s (City of Alexandrina) guidelines, minimum pipe size was adopted as 300 mm for both rainfall events. So, the events of 5% and 10% don’t make much difference at this particular pipe arrangement – but a larger difference would be observed if we made ‘theoretically minimal pipe size’.
The systems are assessed using the two AEP events. I wonder, how the systems could cope with a more extreme event such as an AEP of 1%? Nowadays, it is important to assess the resilience of stormwater systems, see Wang et al., 2019. I would suggest the authors to discuss this aspect of the designed systems. Further, the economic aspect (costs) is only one dimension of the concept of sustainability, see Casal-Campos et al. (2018). It is suggested to discuss this in the context of sustainability including social benefits.According to design guidelines from local government authority (City of Alexandrina), storm events at 10% AEP are considered as minor storms for residential subdivision design purposes, and those at 5% AEP are considered as major storms; typically the 5% AEP allows overland flow to occur within specified limits. Checking the performance of the system when exposed to a much greater storm intensity such as the 1% AEP was considered out of scope with this typical design scenario, but will be considered for future work and this has been noted in Lines 473 to 479 of section 5. Conclusions. As indicated in the answer to Reviewer 3, Question 1, this paper presents only a part of the first author’s PhD project. System performance at extreme event will be investigated and published next.
‘Sustainable Urban Drainage Systems (SUDS)’ should be written as ‘Sustainable Drainage Systems (SuDS)’Thank you for noticing this. It has been corrected in Lines 65-66 of the revised manuscript.
Round 2
Reviewer 1 Report
No substantial modifications were made in the paper, and the issue on novelty remains unsolved. SUDS and drainage systems are widely applied technologies and cost assessments are commonly performed as a part of engineering projects.